# Serum-IgG responses to SARS-CoV-2 after mild and severe COVID-19 infection and analysis of IgG non-responders

Emelie Marklund[1,2], Susannah Leach[3,4], Hannes Axelsson[3], Kristina Nyström[1], Heléne Norder[1,5], Mats Bemark[3,6], Davide Angeletti[3], Anna Lundgren[3,6], Staffan Nilsson[7,8], Lars-Magnus Andersson[1,2], Aylin Yilmaz[1,2], Magnus Lindh[1,5], Jan-Åke Liljeqvist[1,5], Magnus Gisslén[1,2]*

1 Department of Infectious Diseases, Sahlgrenska Academy, University of Gothenburg, Gothenburg, Sweden, 2 Department of Infectious Diseases, Sahlgrenska University Hospital, Gothenburg, Sweden, 3 Department of Microbiology and Immunology, Sahlgrenska Academy, University of Gothenburg, Gothenburg, Sweden, 4 Department of Clinical Pharmacology, Sahlgrenska University Hospital, Gothenburg, Sweden, 5 Department of Clinical Microbiology, Sahlgrenska University Hospital, Gothenburg, Sweden, 6 Department of Clinical Immunology and Transfusion Medicine, Sahlgrenska University Hospital, Gothenburg, Sweden, 7 Department of Mathematical Sciences, Chalmers University of Technology, Gothenburg, Sweden, 8 Department of Laboratory Medicine, Sahlgrenska Academy, University of Gothenburg, Gothenburg, Sweden

* magnus.gisslen@gu.se

**Data Availability Statement:** All files are available from the Zenodo database (DOI: https://doi.org/10.5281/zenodo.3934336).

## Abstract

### Background

To accurately interpret COVID-19 seroprevalence surveys, knowledge of serum-IgG responses to SARS-CoV-2 with a better understanding of patients who do not seroconvert, is imperative. This study aimed to describe serum-IgG responses to SARS-CoV-2 in a cohort of patients with both severe and mild COVID-19, including extended studies of patients who remained seronegative more than 90 days post symptom onset.

### Methods

SARS-CoV-2-specific IgG antibody levels were quantified using two clinically validated and widely used commercial serological assays (Architect, Abbott Laboratories and iFlash 1800, YHLO), detecting antibodies against the spike and nucleocapsid proteins.

### Results

Forty-seven patients (mean age 49 years, 38% female) were included. All (15/15) patients with severe symptoms and 29/32 (90.6%) patients with mild symptoms of COVID-19 developed SARS-CoV-2-specific IgG antibodies in serum. Time to seroconversion was significantly shorter (median 11 vs. 22 days, $P = 0.04$) in patients with severe compared to mild symptoms. Of the three patients without detectable IgG-responses after >90 days, all had detectable virus-neutralizing antibodies and in two, spike-protein receptor binding domain-specific IgG was detected with an in-house assay. Antibody titers were preserved during

**Funding:** This work was supported by the Swedish State Support for Clinical Research (https://www.alfvastragotaland.se, ALFGBG-717531 (MG) and 679621 (SL)) and by SciLifeLab/KAW national COVID-19 research program (https://www.scilifelab.se/covid- 19#nationalprogram, V-2020-0250 (MG) and 2020-0182 (DA, MB, AL)). The funders had no role in study design, data collection and analysis, decision to publish, or preparation of the manuscript.

**Competing interests:** The authors have declared that no competing interests exist.

follow-up and all patients who seroconverted, irrespective of the severity of symptoms, still had detectable IgG levels >75 days post symptom onset.

## Conclusions

Patients with severe COVID-19 both seroconvert earlier and develop higher concentrations of SARS-CoV-2-specific IgG than patients with mild symptoms. Of those patients who not develop detectable IgG antibodies, all have detectable virus-neutralizing antibodies, suggesting immunity. Our results showing that not all COVID-19 patients develop detectable IgG using two validated commercial clinical methods, even over time, are vital for the interpretation of COVID-19 seroprevalence surveys.

## Introduction

The coronavirus disease 2019 (COVID-19) pandemic continues, causing considerable morbidity and mortality worldwide. The severity of COVID-19 ranges from asymptomatic to fatal pneumonitis, with mildly symptomatic patients accounting for approximately 80% of all cases according to current understanding [1]. Severe acute respiratory syndrome coronavirus 2 (SARS-CoV-2), the causative agent of COVID-19, gains entry to human cells by binding the angiotensin-converting enzyme 2 (ACE2) receptor with the receptor-binding domain (RBD) of its spike (S) protein [2]. Thus, antibodies targeting the S-protein may effectively neutralize the virus [3]. Seroprevalence studies usually measure SARS-CoV-2 S- and nucleocapsid (N-) protein specific IgG antibodies; whether these antibodies correlate with protective immunity is however still unknown.

The serological responses to other beta-coronaviruses vary. Whilst all patients infected by SARS-CoV were found to develop IgG antibodies [4], some patients with mild symptoms of MERS-CoV failed to develop detectable levels of IgG [5]. Several short follow-up studies of mostly hospitalized patients have reported the development of IgG in serum against SARS-CoV-2 in the majority of patients [3, 6–10]. When patients remain seronegative, it is often concluded that seroconversion would likely occur later. Serological findings over a longer period than 30 days post symptom onset (PSO) and in patients with non-severe disease remain limited and conflicting. For example, SARS-CoV-2-specific IgG levels in patients have both been found to remain stable approximately 82 days PSO [11] and to wain 2–3 months after infection [7, 12].

Further investigation of patients who fail to produce detectable levels of IgG is lacking and antibody responses in patients with mild symptoms are also poorly described. Here, we investigated serum-IgG (S-IgG) responses to SARS-CoV-2 in a cohort of patients with both severe and mild COVID-19, profiling the patients who remained seronegative.

## Materials and methods

### Patients and sample collection

A cohort of 47 patients were recruited between February 25th and March 25th 2020, at the Department of Infectious Diseases, Sahlgrenska University Hospital, Gothenburg, Sweden. The study protocol was approved by the Swedish Ethical Review Authority (Registration number 2020–01771) and patients were included after written informed consent. Disease severity was divided into severe and mild: severe cases were defined as those requiring invasive mechanical ventilation or high-flow nasal oxygen, and mild cases as not requiring oxygen nor

in-patient hospital care [13]. Blood samples were collected during hospitalization and/or during follow-ups. Seroconversion was defined as detectable levels of SARS-CoV-2-specific IgG antibodies in serum.

## Real-Time Polymerase Chain Reaction (RT-PCR) assay

All patients had been diagnosed with SARS-CoV-2 with RT-PCR from the upper respiratory tract (pooled nasopharyngeal and throat swabs) during acute phase of the infection. Nucleic acid was extracted from clinical samples in a MagNA Pure 96 instrument using the Total Nucleic Acid isolation kit (Roche). RT-PCR targeting the RdRP region was performed in a QuantStudio 6 instrument (Applied Biosystems, Foster City, CA) using the primers and probe described [14]. Cycle threshold (Ct) values <38 were regarded as positive.

## Protein expression

The SARS-CoV2 spike protein RBD (amino acids 319–541) was produced using an expression vector obtained through BEI Resources, NIAID, NIH, which is vector pCAGGS containing the SARS-CoV-2, Wuhan-Hu-1 spike glycoprotein gene RBD with C-terminal Hexa-Histidine tag (NR-52309). 293F cells (Cat nr R79007, ThermoFisher Scientific) were cultured in Freestyle 293 medium at 37˚C in 5% $CO_2$ in Optimum Growth$^{TM}$ flasks (Thomson instrument company) at 130 rpm in a Multitron 4 incubator (Infors) and transfected at 2xE6 cells/ml using FectoPro transfection reagent (Polyplus transfection). Protein-containing culture supernatant (1L) was harvested after 90h, filtered using Polydisc AS 0.45 µm (Whatman) and loaded onto a 5 mL HisExcel column (GE healthcare). The column was washed with 20 mM sodium phosphate, 0.5M NaCl and 30 mM imidazole before elution of the protein using the same buffer but with 300 mM imidazole. Pooled fractions were concentrated using 10 kDa Vivaspin concentrators (MWCO 10 kDa, Sartorius), passed over a HiPrep 26/10 desalting column (GE Healthcare) in phosphate-buffered saline and finally concentrated again.

## Detection of SARS-CoV-2 specific serum antibodies

Serum-IgG antibodies against SARS-CoV-2 were analyzed using two commercially available serological assays: the qualitative Architect chemiluminescent microparticle immunoassay (Abbott Laboratories, USA), measuring IgG against SARS-CoV-2 N-protein, and the quantitative iFlash 1800 chemiluminescent immunoassay (YHLO, China), which measures IgG against both SARS-CoV-2 S- and N-proteins. All samples were analyzed using both assays. IgG concentrations were obtained using the iFlash 1800 assay, and ≥10 AU/ml were defined as positive. The time of seroconversion was defined as the time-point at which the first positive serum-IgG result was observed.

The RBD ELISA was performed as previously described [15] with some modification. Plates were coated with 25µL per well at 4µg/mL SARS-CoV2 RBD in 1X PBS, incubated overnight at 4˚C. Blocking was done with 100µL per well of 2% sterile filtered BSA in 1X PBS at RT for 1hr. Plates were washed with 1X PBS + 0.05% Tween using a 405 LS Washer, Biotek and incubated with 25µL sera (not heat-inactivated) per well for 1hr, with a starting dilution of 1:50 followed by 2-fold serial dilution. Secondary antibodies were diluted in 1X PBS + 0.05% Tween all 1:6000 and 25µL added to each well at RT for 1hr. Antibodies used: Goat Anti-Human IgG Fc-HRP (SouthernBiotech, cat. No. 2048–05), Goat Anti-Human IgA-HRP (SouthernBiotech, cat. No. 2050–05), Goat Anti-Human IgM-HRP (SouthernBiotech, cat. No. 2020–05). After washing, signal was developed with addition of 1-step Ultra TMB-ELISA (ThermoFisher, cat. No. 34029) and reaction was stopped after 5 min by addition of 2M $H_2SO_4$. OD for plates was measured at 450nm. AUC was calculated in GraphPad Prism 8 (GraphPad Software). The

assay was verified by using 10 negative samples, collected prior to COVID-19 outbreak and 10 positive samples from patients hospitalized for COVID-19 infection. Seropositivity was defined as AUC value over mean+2SD of negative samples.

Total concentrations of Ig A, G and M were determined using commercially available reagents on the Alinity platform (Abbott Laboratories).

### Flow cytometry

Blood was collected in EDTA Vacutainer tubes and was analyzed using Multitest 6-Color TBNK reagent (337166) or anti-CD45-PerCP (345809)/anti-CD14-APC (345787) in Trucount tubes (340334) according to normal procedures using FACSCanto II flow cytometers (all from BD Biosciences, San Jose, CA).

### SARS-CoV-2 Neutralizing antibody assay

Neutralizing antibodies (NAb) were determined after inactivation of the complement in serum for 30 minutes at 56˚C, by incubating 25μL of 2-fold dilutions (1/2–1/264) of each serum in maintenance medium (MM) with 25 μL of $100TCID_{50}$ of SARS-CoV-2 (DE strain, isolated from sample collected February 25, 2020) in duplicate for two hours at 37˚C. Thereafter the serum/virus mixture was added to confluent Vero cells (ATCC CCL-81) in 96 well microtiter plates with 175 μL MM and incubated at 37˚C in a $CO_2$ incubator. Ten-fold serial dilutions of the virus, 10–1,000 $TCID_{50}$, were added in duplicate to separate wells as an infection control. The plates were examined daily, using an inverted microscope, and complete cytopathic effect (CPE) was usually complete in the virus only control-wells after three days. The presence of any CPE of the cells was then recorded in the wells and the titer of the sera was calculated as previously described [16]. Sera with antibody titers ≥4 (1/4 dilution) were considered neutralizing, demonstrating neutralization in both 1/2 and 1/4 dilutions, confirming the presence of antibodies with capacity to block infection. Sera (n = 17) from patients and blood donors from before December 2019 were used as a negative control and no neutralization was detected in any of these samples.

### Statistical analysis

Time to event analysis with interval censoring was used to compare time to seroconversion between groups [17]. Differences between groups were analyzed using Welch's t-test and longitudinal changes with paired t-test. P<0.05 was considered statistically significant. Statistical analyses were performed using GraphPad Prism 8 (GraphPad Software, Inc) and R 4.0.0.

## Results

### SARS-CoV-2-specific serum-IgG antibodies in severe and mild COVID-19

Forty-seven patients provided a total of 156 serum samples (mean 3.3 per patient, range 1–7), 5–117 days PSO. Of the 47 patients, 15/47 (32%) had severe and 32/47 (68%) had mild COVID-19 (Table 1). The patients with severe symptoms were older (mean age 58) and all male, compared to patients with mild symptoms (mean age 45, 56% female).

All 15 patients with severe COVID-19 developed SARS-CoV-2-specific IgG antibodies in serum in both of the commercial IgG assays (Architect and iFlash). Of the patients with mild symptoms, 29/32 (90.6%) developed SARS-CoV-2-specific IgG antibodies. Among these, 27/29 developed detectable IgG antibodies in both of the commercial IgG assays (Architect and iFlash) and 2/29 were classified as negative (index <1.4) in Architect, although the index values were clearly above negative samples (index between 0.83–0.95). Follow-up samples from

**Table 1. Demographic and clinical characteristics of 47 patients with COVID-19.**

|  | Mild n = 32 | Severe n = 15 | Total n = 47 |
|---|---|---|---|
| *Characteristics* |  |  |  |
| Age, mean (range) | 45 (19–71) | 58 (46–81) | 49 (19–81) |
| Female, no. (%) | 18 (56) | 0 (0) | 18 (38) |
| *Max level of care* |  |  |  |
| Outpatient, no. (%) | 32 (100)[a] | 0 (0) | 32 (68) |
| Hospitalized, no. (%) | 0 (0) | 15 (100) | 15 (32) |
| ICU, no. (%) | 0 (0) | 12 (80) | 12 (26) |
| *Comorbidities* |  |  |  |
| Hypertension, no. (%) | 1 (3) | 5 (33) | 4 (9) |
| Diabetes, no. (%) | 0 (0) | 2 (13) | 2 (4) |
| Heart Disease, no. (%) | 1 (3) | 2 (13) | 3 (6) |
| Cerebrovascular disease, no. (%) | 0 (0) | 1 (7) | 1 (2) |
| Asthma, no. (%) | 3 (9) | 1 (7) | 4 (9) |

[a]7 patients in this group were isolated in hospital to prevent viral transmission according to control policy at that time but did not require hospital care. ICU = Intensive Care Unit.

these two patients presented similar reactivity in Architect which could be considered as grey-zone reactivity and as all samples were positive in iFlash, these two patients were interpreted as IgG-positive.

Among the 15 patients with severe symptoms, seroconversion was observed after in median 11 (range 7–20) days PSO (Fig 1A). Among the 29/32 patient with mild symptoms that were considered IgG positive, seroconversion was observed after median 22 (range 14–79) days PSO (Fig 1B). When accounting for the varying sampling times by using interval censoring, time to seroconversion was still significantly shorter in patients with severe symptoms than in those with mild symptoms (P = 0.04). Furthermore, we found significantly higher concentrations of IgG antibodies in patients with severe symptoms (mean 107 AU/ml) than in patients with mild symptoms (mean 65 AU/ml) within 35 days PSO (P = 0.004; Fig 1C). Within both groups, antibody concentrations did not change significantly in patients >75 days, hence the differences between the groups remained (though no longer significantly (P = 0.294, Fig 1C) due to fewer patients analyzed at this time-point).

Among patients with mild symptoms, 3/32 (9.4%) did not develop detectable IgG antibodies as determined using the commercially available assays during the follow-up period, 91–105 days PSO.

## Analysis of patients without detectable IgG-responses

We assessed the three patients without detectable IgG-responses further (Table 2, Fig 2). The lowest observed Ct values in the non-IgG patients were of a similar range as in the patients with mild symptoms who seroconverted (15.9–24.9 vs. 11.9–37.2). Additionally, the non-IgG patients had similar range of duration of symptoms (20–28 days) as the patients with mild symptoms who seroconverted (1–44 days, data from 26/29 patients). Flow cytometric analyses of peripheral blood cells in two of the non-seroconverting patients available for additional testing (days 91 and 105 PSO) revealed no overt signs of immunodeficiency (Table 3). Total concentrations of IgG, IgA and IgM isotypes in serum sampled at the same time-points were also within the normal adult range, indicating no antibody deficiencies (Table 2).

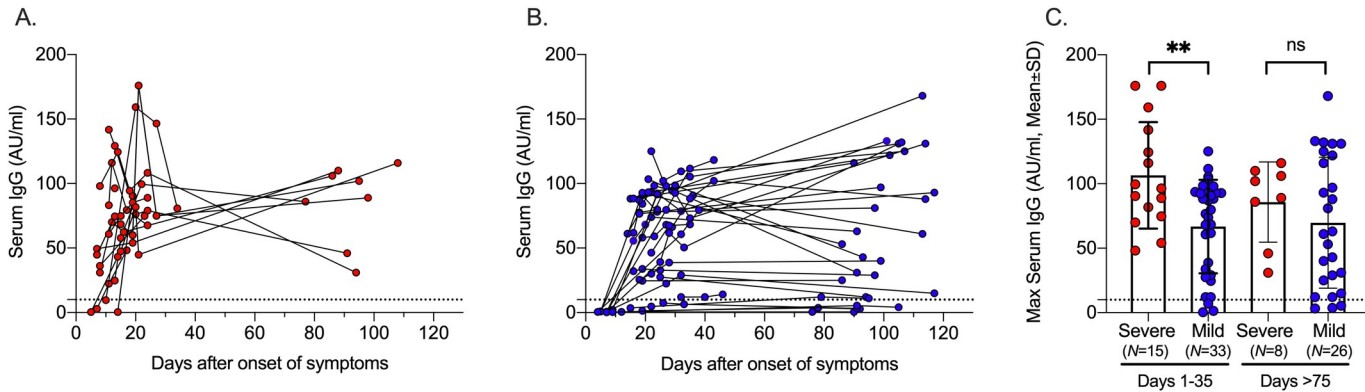

**Fig 1. SARS-CoV-2-specific serum IgG antibody responses in patients with severe and mild COVID-19.** Concentrations of serum IgG (AU/ml) over time in patients with severe (A, red) and mild (B, blue) disease. (C) Maximum concentrations of serum IgG (AU/ml) in early (1–35 days) and late (>75 days) follow-up post symptom onset. Cut-off for positive sample indicated by dotted line. ns P>0.05, ** P<0.01.

## Neutralizing and RBD-specific antibodies in patients without detectable IgG-responses

To further investigate the humoral immune response against SARS-CoV-2 in the three patients without detectable IgG, neutralizing antibodies, considered the golden standard of anti-viral serological testing, were analyzed in serum samples collected days 78–91 PSO. All 3 patients had detectable NAb activity (Table 2) indicating that these patients had mounted a functional humoral immune response against SARS-CoV-2. To ascertain if the neutralizing activity could be explained by low antibody levels against the RBD undetected by the commercial assays, an in-house ELISA was used (Table 2, Fig 3). Indeed, the two patients with highest levels of neutralization had measurable anti-RBD-IgG in the same samples whilst the patient

**Table 2. Demographic, clinical and laboratory findings in three patients with undetectable levels of serum IgG against SARS-CoV-2 after COVID-19 using commercially available kits.**

|  | Patient 1 |  | Patient 2 |  | Patient 3 |  |
|---|---|---|---|---|---|---|
| **Age** | **46** |  | **19** |  | **43** |  |
| Sex | Male |  | Female |  | Female |  |
| No. symptomatic days | 28 |  | 29 |  | 20 |  |
| Lowest Ct-value[a] | 23 |  | 24.9 |  | 15.9 |  |
| No. positive viral PCR | 6 |  | 3 |  | 3 |  |
| Total serum antibody concentrations[b] | IgG | 8.1 | IgG | 10 | IgG | 9.7 |
|  | IgA | 1.8 | IgA | 1.2 | IgA | 0.9 |
|  | IgM | 0.48 | IgM | 1.4 | IgM | 1.5 |
| α-SARS-CoV-2 IgG | Neg |  | Neg |  | Neg |  |
| Neutralizing ab titer[c] | 12 |  | 8 |  | 48 |  |
| α-RBD antibodies | IgG | Pos | IgG | Neg | IgG | Pos |
|  | IgA | Neg | IgA | Neg | IgA | Neg |
|  | IgM | Neg | IgM | Neg | IgM | Neg |

Blood samples analyzed for specific antibodies and total serum antibodies collected at day 76 (patient 2) and 91 (patient 1 and 3) post symptom onset.

[a]Ct values <38 are considered positive.

[b]Normal range: IgM 0.27–2.1, IgG 6.7–15, IgA 0.88–4.5 g/L. RDB = Receptor binding domain.

[c]Titers ≥4 are considered positive.

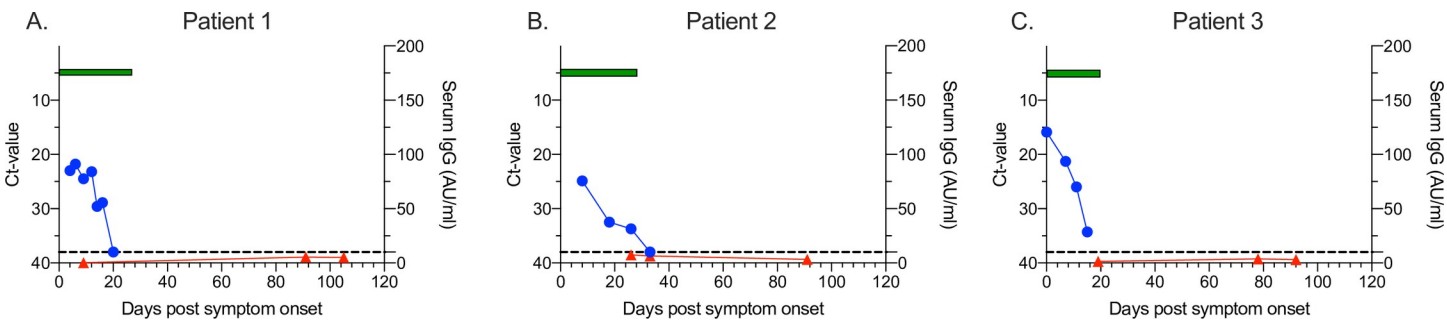

**Fig 2. SARS-CoV-2 viral load, IgG antibody concentration and symptom duration in three IgG-negative patients >90 days post onset of symptoms.** Ct values over time (blue circles, left y-axis), concentration of SARS-CoV-2-specific serum IgG antibodies over time (red triangles, right y-axis) and number of days of with symptoms (green bar, x-axis). Cut-off for positive viral sample indicated by dotted line. ns P>0.05, * P<0.05, ** P<0.01, *** P<0.001.

with low neutralizing activity lacked detectable anti-RBD-IgG (Table 2). Compared to late serum samples (>75 days PSO, n = 20) from patients with detectable anti-SARS-CoV-2-IgG with mild and severe disease, these three patients had lower levels of anti-RBD-IgG, IgA and IgM (Fig 3).

## Discussion

In this study, we describe IgG antibody responses in 47 patients during and after severe and mild COVID-19. Patients with severe disease seroconverted earlier and had higher maximum concentrations of anti-viral IgG than those with mild disease. Whilst all of the patients with severe symptoms seroconverted, three (9%) of the 32 patients with mild disease failed to produce levels of IgG detectable with commercial assays, even more than 90 days PSO. However, NAbs against SARS-CoV-2 were detected in all of these three patients.

Our results confirm previous findings that clinical severity of disease is associated with higher SARS-CoV-2-specific serum-IgG antibodies [18–21]. Studies comparing time to seroconversion between these groups are still lacking and we show that clinical severity is also associated with significantly earlier seroconversion. Due to the long follow-up period of this study, we were also able to observe that all seroconverted patients with both mild and severe symptoms still have detectable IgG levels after more than 75 days. Long et al reported that 97% of 37 patients with mild COVID-19 had decreased levels of IgG 2–3 months PSO [7], while another study with 34 hospitalized patients with COVID-19 presented increased levels of IgG until 5 weeks PSO, followed by consistent levels up to 7 weeks PSO [22]. Interestingly, our study shows that several patients with both mild and severe symptoms had increased in IgG concentrations over time. Wajnberg et al have shown that S-protein IgG levels could increase up to mean 82 days PSO in

**Table 3. Flow cytometric analysis of peripheral blood cells in two patients with undetectable levels of serum IgG against SARS-CoV-2 using commercially available kits after COVID-19.**

| Cell type | Phenotype | Patient 1[a] | Patient 3 | Normal range |
|---|---|---|---|---|
| T cells | CD3+ | 1,1 | 2 | 0,7–2,1 |
| T helper cells | CD3+4+ | 0,69 | 1,19 | 0,3–1,4 |
| T cytotoxic cells | CD3+8+ | 0,35 | 0,66 | 0,2–0,9 |
| B cells | CD19 | 0,08 | 0,25 | 0,1–0,5 |
| NK cells | CD3-56+ | 0,12 | 0,15 | 0,09–0,6 |
| Classical monocytes | CD14+16- | 0,17 | 0,31 | 0,26–0,38 |

[a]Samples collected at 105 (patient 1) and day 91 (patient 3) post symptom onset.

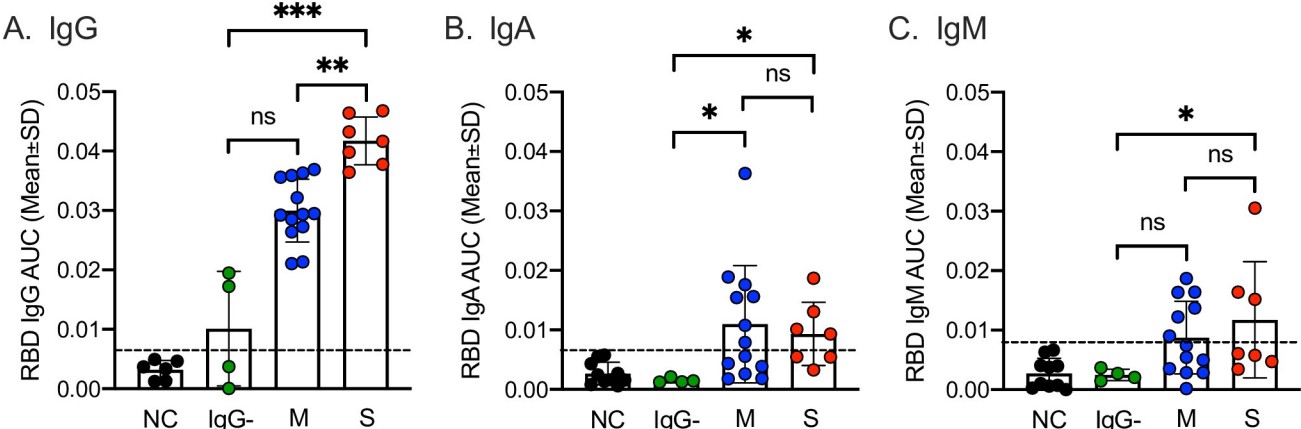

**Fig 3. Antibody responses against the receptor-binding domain of SARS-CoV-2.** RBD-specific serum IgG (A), IgA (B) and IgM (C) antibodies in patients with severe symptoms (red, n = 7), mild symptoms and IgG-positive (blue, n = 13), mild symptoms and IgG-negative (green, n = 3) collected 78–91 days post symptom onset. NC = negative controls (black dots, n = 10). Cut-off (mean + 2SD of NC AUC) indicated by dashed lines. ns P>0.05, * P<0.05, ** P<0.01.

patients whose antibody levels were low initially [11]. The discrepancies between the results may at least partly be explained by different target antigens used in antibody detection in the different studies. Of note, all of the patients in the severe group in this study were male. In a study comparing serum-IgG responses between the sexes, no significant differences were found between women and men with severe COVID-19 [23], suggesting that the differences in proportions of males within the groups are unlikely to explain the concentration differences.

In this study, we found that almost 10% of patients with mild COVID-19 did not develop detectable anti-SARS-CoV-2 IgG in serum as evaluated by assays used in clinical practice. Previous studies have similarly failed to detect IgG antibodies in patients with mild disease [24–26], but due to short follow-up (less than 25–50 days) no conclusions regarding the proportion of patients who do not seroconvert have been made in the belief that antibody levels become detectable later in time. We show that despite 90 days or more PSO, not all patients develop detectable levels of IgG in these assays. However, using a virus neutralization assay, considered the golden standard of serology testing, all patients with undetectable IgG using commercial methods had NAbs. Furthermore, when using an in-house RBD ELISA, the two patients with highest levels of NAbs also had detectable anti-RBD-IgG. Several other studies have seen a correlation between the magnitudes of NAbs and IgG against viral S-protein epitopes [10, 27]. These results suggest that 100% of patients had in fact seroconverted, compared to 91% detected when using commercial methods.

While none of the patients with undetectable IgG using the commercial assays had anti-RBD-IgA, interestingly, the patients with detectable IgG also had detectable anti-RBD-IgA >75 days PSO, with no difference in IgA levels between mild and severe cases. Other studies have found serum-IgA after mild COVID-19 to be transient and undetectable after only one month post recovery [18, 28], again highlighting the sensitivity of our in-house RBD-ELISA. It is unsurprising that the validated SARS-CoV-2 antibody assays used in clinical practice miss a proportion of positive samples which other assays may detect. In these situations, the balance between assay sensitivity and specificity must be weighted to reduce the risk of false positives. This study demonstrates the risk of false negatives, in which low serum antibody levels may be detected with more sensitive assays.

We acknowledge several limitations in this study. The number and timing of serum sampling differed between patients; detectable IgG may have occurred before sampling, meaning

that the exact timing of seroconversion is uncertain. Also, a larger proportion of patients in the mild group than the severe group have been available for late follow-up sampling.

## Conclusions

We show that patients with severe COVID-19 both seroconvert earlier and develop higher concentrations of SARS-CoV-2-specific IgG than patients with mild symptoms. That not all COVID-19 patients develop detectable levels of IgG using two validated commercial methods, even over time, are vital for the interpretation of COVID-19 seroprevalence surveys and estimating the true prevalence in populations.

## Acknowledgments

We thank the Mammalian Protein core facility (MPE) at the University of Gothenburg for protein production.

## Author Contributions

**Conceptualization:** Heléne Norder, Lars-Magnus Andersson, Aylin Yilmaz, Magnus Gisslén.

**Data curation:** Emelie Marklund, Magnus Gisslén.

**Formal analysis:** Emelie Marklund, Susannah Leach, Hannes Axelsson, Staffan Nilsson, Magnus Lindh, Magnus Gisslén.

**Funding acquisition:** Mats Bemark, Davide Angeletti, Anna Lundgren, Magnus Gisslén.

**Investigation:** Emelie Marklund, Susannah Leach, Hannes Axelsson, Kristina Nyström, Heléne Norder, Mats Bemark, Davide Angeletti, Anna Lundgren, Magnus Lindh, Jan-Åke Liljeqvist, Magnus Gisslén.

**Methodology:** Emelie Marklund, Susannah Leach, Hannes Axelsson, Kristina Nyström, Heléne Norder, Mats Bemark, Davide Angeletti, Anna Lundgren, Staffan Nilsson, Magnus Lindh, Jan-Åke Liljeqvist, Magnus Gisslén.

**Project administration:** Emelie Marklund, Magnus Gisslén.

**Resources:** Mats Bemark, Davide Angeletti, Anna Lundgren, Lars-Magnus Andersson, Aylin Yilmaz, Magnus Lindh, Jan-Åke Liljeqvist, Magnus Gisslén.

**Supervision:** Susannah Leach, Magnus Gisslén.

**Validation:** Emelie Marklund, Susannah Leach, Magnus Gisslén.

**Visualization:** Emelie Marklund, Susannah Leach, Hannes Axelsson, Mats Bemark, Magnus Gisslén.

**Writing – original draft:** Emelie Marklund, Susannah Leach, Magnus Gisslén.

**Writing – review & editing:** Hannes Axelsson, Kristina Nyström, Heléne Norder, Mats Bemark, Davide Angeletti, Anna Lundgren, Staffan Nilsson, Lars-Magnus Andersson, Aylin Yilmaz, Magnus Lindh, Jan-Åke Liljeqvist.

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
