## [Decision Letter · Decision Letter 0]

3 Aug 2020

PONE-D-20-21112

Serum-IgG responses to SARS-CoV-2 after mild and severe COVID-19 infection and analysis of IgG non-responders

PLOS ONE

Dear Dr. Gisslén,

Thank you for submitting your manuscript to PLOS ONE. After careful consideration, we feel that it has merit but does not fully meet PLOS ONE’s publication criteria as it currently stands. Therefore, we invite you to submit a revised version of the manuscript that addresses the points raised during the review process.

We look forward to receiving your revised manuscript.

Kind regards,

Stephen R. Walsh, MDCM

Academic Editor

PLOS ONE

Journal Requirements:

2.PLOS ONE requires experimental methods to be described in enough detail to allow suitably skilled investigators to fully replicate and evaluate your study. See https://journals.plos.org/plosone/s/submission-guidelines#loc-materials-and-methods for more information. To comply with PLOS ONE submission guidelines, in your Methods section, please provide a more detailed description of how you produced SARS-CoV2 RBD.

In addition, if methods are well established, authors may cite articles where those protocols are described in detail, but the submission should include sufficient information to be understood independent of these references. Please revise your manuscript so that cited protocols are briefly but sufficiently described.

3. In your Methods section, please provide additional details regarding the 293F and Vero cell lines used in your study. Please include the source from which you obtained the cells, the catalog number if applicable, whether the cell line was verified, and if so, how it was verified. For more information on PLOS ONE's guidelines for research using cell lines, see https://journals.plos.org/plosone/s/submission-guidelines#loc-cell-lines.

4.We note that you have stated that you will provide repository information for your data at acceptance. Should your manuscript be accepted for publication, we will hold it until you provide the relevant accession numbers or DOIs necessary to access your data. If you wish to make changes to your Data Availability statement, please describe these changes in your cover letter and we will update your Data Availability statement to reflect the information you provide.

5.PLOS requires an ORCID iD for the corresponding author in Editorial Manager on papers submitted after December 6th, 2016. Please ensure that you have an ORCID iD and that it is validated in Editorial Manager. To do this, go to ‘Update my Information’ (in the upper left-hand corner of the main menu), and click on the Fetch/Validate link next to the ORCID field. This will take you to the ORCID site and allow you to create a new iD or authenticate a pre-existing iD in Editorial Manager. Please see the following video for instructions on linking an ORCID iD to your Editorial Manager account: https://www.youtube.com/watch?v=_xcclfuvtxQ

<h1>** **</h1>

Additional Editor Comments (if provided):

I would like to thank the authors for choosing to submit their manuscript for consideration by PLoS One. We have sought the opinion of three peer reviewers who are experts in the field and they have raised a number of issues. In their opinion, and mine, the manuscript could be considerably improved by addressing each of these issues. If the authors wish to address these comments and suggestions, please respond in a point-by-point fashion to each one.

Reviewers' comments:

Reviewer's Responses to Questions

**Comments to the Author**

1. Is the manuscript technically sound, and do the data support the conclusions?

Reviewer #1: Partly

Reviewer #2: Partly

Reviewer #3: Yes

2. Has the statistical analysis been performed appropriately and rigorously? 

Reviewer #1: Yes

Reviewer #2: I Don't Know

Reviewer #3: Yes

3. Have the authors made all data underlying the findings in their manuscript fully available?

Reviewer #1: Yes

Reviewer #2: No

Reviewer #3: Yes

4. Is the manuscript presented in an intelligible fashion and written in standard English?

Reviewer #1: Yes

Reviewer #2: Yes

Reviewer #3: Yes

5. Review Comments to the Author

Reviewer #1: The manuscript by Marklund et al describes the analysis of serum IgG responses to SARS-CoV-2 in a cohort of patients with severe (n=15) or mild (n=32) COVID-19 disease. Patients with severe disease were found to seroconvert earlier and had higher IgG concentrations within the first 35 days post-symptom onset compared to patients with mild disease. It was noted that three individuals with mild disease never demonstrated detectable IgG responses using two commercially available serological assays. A virus neutralization assay was used to demonstrate that detectable serum neutralizing antibody activity could be detected in all three of these individuals, and two patients also had detectable IgG responses to RBD using an in-house ELISA assay. The authors conclude that commercially available assays may not have the sensitivity or specificity to detect positive IgG antibody responses in patients with mild disease, which will be important for interpreting results from seroprevalence studies. Overall the finding that commercial kits may have a rate of false negative results is not surprising, as even the authors state in the discussion. Additional data would be helpful in interpreting the antibody responses in these three individuals and comparisons with other individuals in the mild disease cohort that did develop positive serum reactivity. Below are more specific comments for consideration.

1: Materials and Methods. The description of the neutralizing antibody assay is confusing. Serial dilutions of serum are co-incubated with a standardized amount of virus for 2 hours at 37’C, and then the mixture added to Vero target cells, as per standard neutralizing antibody assay protocol. But then it describes adding ten-fold serial dilutions of virus. Is this for separate wells, or the positive control? What are the criteria for determining a serum dilution is “positive” for neutralizing activity? It mentions measuring CPE, but not clear if plaque reduction is being measured? Were normal human serum samples used as negative control to determine background of the assay? Clear details regarding this assay should be provided.

2: The authors state that all serum samples were tested using two commercially available serological assays, but data are only presented for one of the assays. Was there general concordance between the two assays or were discrepancies noted? This should be discussed, and data from the second assay platform should be presented.

3: It is difficult to interpret the data presented in Figure 2 without a similar comparison of individuals with mild disease that did develop IgG responses. It is not clear if there could be differences in Ct values over time or number of days with symptoms between the two categories of mild disease patients that may help characterize the development of the antibody response. Figure 2 would be more informative if the same parameters were graphed for patients with mild disease with detectable IgG responses.

4: Neutralizing antibody titers were only measured at a single late timepoint for the three individuals without detectable IgG. The titers measured are very low, so it would be helpful to have data from negative control serum samples to demonstrate background of the assay and validate that the responses measured are real. It would also be informative to understand the neutralizing antibody responses at earlier timepoints closer to PSO, as perhaps they would be more robust. Likewise, it would be helpful to understand the titers of neutralizing antibodies measured in mild patients that do develop detectable IgG titers (Table 2) for comparison.

5: Figure 1C should include labels to differentiate the two sets of scatter plots (i.e. Days 1-35 and >75 days).

6: Table 2: Annotations for “b” and “c” are mixed up in table legend.

Reviewer #2: Reviewer Summary:

The main claims of the paper are that anti-SARS-COV-2 IgG responses may be detected by a neutralizing antibody assay or sensitive ELISA in instances of mild infection where these responses were not detected in commercially available serology assays. Of interest are the findings that patients with severe symptoms tend to seroconvert faster than those with milder symptoms, and it seems that these antibody responses do not decrease over time (in the first 90 days or so), in contrast to the results of other studies with SARS-CoV-2 and other beta coronaviruses. These findings have implications for the interpretation of serology assays when used in seroprevalence surveys. This paper confirms the existence of the wide range in sensitivity of various serology assays (both commercial and lab-based), and raises an interesting point that serum neutralization may be a more sensitive readout than binding to the SARS-CoV-2 spike protein or the RBD of the spike protein. The manuscript is well organized and written clearly overall, but additional detail could be added to help non-specialists to understand the data and its interpretation. There are many opportunities available to add detail to the methods, results, and discussion sections to clarify to strengthen the interpretation of the data and to provide a resource for others looking to expound upon these results with other SARS-CoV-2 infection cohorts. Overall the manuscript has good potential for publication and these additional details would strengthen the manuscript so that it would be fit for publication.

Specific Areas for Improvement:

General: Timelines of antibody seroconversions are referred to frequently in the manuscript without context – e.g. “11 vs. 22 days”. Does this refer to post symptom onset? This should be clarified throughout.

Abstract: In the last sentence of the abstract, it is stated that “Our results…are vital for…. estimating the true infection prevalence in populations.” Is it true that antibody serology assays are being used to measure both seroprevalence AND infection prevalence? It is my understanding that RT-PCR is being used in most cases to measure infection prevalence.

Introduction: Literature on SARS-CoV-2 serology is constantly increasing and our thinking is changing by the day. It would be worthwhile to add more context to previous literature, including what has emerged in the past few weeks since submission if possible.

Methods: I could not find any supplemental information for this manuscript and thus no detailed protocols. The authors do refer to the Amanat et al paper for their SARS-CoV-2 ELISA and it appears that they made some modifications, but more details and a rationale for changing the conditions would be very helpful to the reader. It would be helpful to add more details for the RBD ELISA, such as incubation times, what positive controls were used, the readout (concentration determined by normalizing to a standard curve or OD) and how the threshold for a positive response was determined. There were also several modifications done compared to the published method by Amanat et al, and it would be helpful to provide a rationale for these differences (such as the increased coating concentration, differences in buffers/diluents, etc). It is also important to state whether the samples were also heat inactivated before the analysis of binding antibodies (either with the commercial assays or the RBD ELISA). Please state what positive controls were used in the RBD ELISA and how the threshold for a positive response was calculated. For the neutralization assay, which strain of SARS-CoV-2 was used, and what were the controls in the assay? How was positivity determine in the neutralization assay?

I do not have extensive statistical expertise to comment on the methods used, but I have raised some questions regarding Figure 3 (below) that could be added to the section on statistical analysis.

Results and Data: Overall the results section tells a nice story and it easy to follow from one figure/table to the next. There are some details that could be added that would help the reader better understand and interpret the results for figure 3.

Data could not be found in a repository. The description of data location is “All files are available from the Zenodo database (DOI:XXX)” so I was unable to find this data.

Table 2: the footnotes for b and c are switched.

Figure 3: What p values are denoted by the * and **? It would also be helpful to have a positivity cutoff line, similar to what you have in Figure 1 as well as an explanation of how the positivity threshold was calculated. What percentage of samples in the Mild and Severe infection categories would be positive? How is AUC calculated and can this be added to the methods/stats section? What is the readout of the RBD ELISA- concentration or OD? Is it possible that the low titers of IgA and IgM are due to competition with high levels of IgG for binding to the RBD?

Discussion: It is intriguing that the authors claim they did not observe a notable decline in antibody responses with the commercial assays. This should be discussed in further detail in the context of current literature on SARS-CoV-2. Would the same results be expected if a longitudinal analysis was conducted using the RBD ELISA or the neutralization assay? Also, on page 12 line 277, the authors refer to the increased sensitivity of their in-house ELISA. Is this more sensitive than the one published by Amanat et al? Did the various modifications make the ELISA more sensitive than the one published by Amanat et al? Or is the statement just meant to compare the RBD ELISA to the commercial serology assays? Clarifying and expanding on these points would further support your claims and be benefit to the field of SARS-CoV-2 serology.

Reviewer #3: The manuscript described a series of covid patients and revealed that patients with mild symptoms developed antibodies with lower titers than those with severe symptoms. The authors also identified three patients with undetectable antibody titers by commercial IgG assays. The study has certain merits but with some issues:

1. In abstract, there is no methods section.

2. For the RBD in-house antibody assay, how to decide the cut-off should be stated in the methods.

3. For neutralizing antibody assay, how to decide the cut-off should be clarified in the methods.

4. In Line 167, a p-value is needed.

5. Missing data from patient2 in Table 3 and Table 3 should be presented as supplemental table.

6. In Line 273-282, the argument about anti-RBD-IgA may not be true. One sample in the M group has a very low IgA (the dot is touching the 0.00 base line in Figure 3). The high sensitivity of in-house assay may come with high false positivity. IgA covid tests are known for their high false positivity.

6. PLOS authors have the option to publish the peer review history of their article (what does this mean?). If published, this will include your full peer review and any attached files.

Reviewer #1: No

Reviewer #2: No

Reviewer #3: No

---

## [Author Response · Author response to Decision Letter 0]

1 Sep 2020

Page and line numbers refer to the Revised manuscript (clean copy).

Journal requirements

1. To comply with PLOS ONE submission guidelines, in your Methods section, please provide a more detailed description of how you produced SARS-CoV2 RBD.

A detailed description of the protein expression has been added the methods section (page 4, lines 97-110).

2. In your Methods section, please provide additional details regarding the 293F and Vero cell lines used in your study. Please include the source from which you obtained the cells, the catalog number if applicable, whether the cell line was verified, and if so, how it was verified. 

These details have now been added (page 5, line 100 and page 7, line 150). Both cell lines were bought commercially and have not been further verified.

3. We note that you have stated that you will provide repository information for your data at acceptance. Should your manuscript be accepted for publication, we will hold it until you provide the relevant accession numbers or DOIs necessary to access your data.

The DOI is https://doi.org/10.5281/zenodo.3934336, which has been added to the relevant section in Editorial Manager.

4. PLOS requires an ORCID iD for the corresponding author in Editorial Manager.

Magnus Gisslén (the corresponding author) ORCID iD: 0000-0002-2357-1020.

We have not been able to enter it into Editorial Manager due to multiple accounts, but a merge is in process.

Reviewer #1

1: Materials and Methods. The description of the neutralizing antibody assay is confusing. Serial dilutions of serum are co-incubated with a standardized amount of virus for 2 hours at 37’C, and then the mixture added to Vero target cells, as per standard neutralizing antibody assay protocol. But then it describes adding ten-fold serial dilutions of virus. Is this for separate wells, or the positive control? What are the criteria for determining a serum dilution is “positive” for neutralizing activity? It mentions measuring CPE, but not clear if plaque reduction is being measured? Were normal human serum samples used as negative control to determine background of the assay? Clear details regarding this assay should be provided.

The method description for the neutralizing antibody assay has now been adjusted in accordance with the reviewer’s comments (Page 6, lines 146-160).

2: The authors state that all serum samples were tested using two commercially available serological assays, but data are only presented for one of the assays. Was there general concordance between the two assays or were discrepancies noted? This should be discussed, and data from the second assay platform should be presented.

There was general concordance between the two assays, with a discrepancy seen in only two patients. This is now presented clearly in the results section (page 8, lines 188-193). Since the Architect assay is qualitative and not quantitative, we see no additional value in presenting the results from that assay separately, except for the two patients mentioned above. 

3: It is difficult to interpret the data presented in Figure 2 without a similar comparison of individuals with mild disease that did develop IgG responses. It is not clear if there could be differences in Ct values over time or number of days with symptoms between the two categories of mild disease patients that may help characterize the development of the antibody response. Figure 2 would be more informative if the same parameters were graphed for patients with mild disease with detectable IgG responses.

It is unfortunately not feasible to graph the same parameters for the 29 patients with mild disease and detectable IgG responses. Instead, we have added details of these patients in the results section (page 10, lines 218-222). We state that the lowest observed Ct-values in the non-IgG patients were of a similar range as in the patients who seroconverted (15.9–24.9 vs. 11.9–37.2), and it has now been clarified that this is referring to the seroconverted patients with mild disease. Further, durations of symptoms in the non-IgG patients and in the majority of the patients with mild symptoms who seroconverted has been added. 

4: Neutralizing antibody titers were only measured at a single late timepoint for the three individuals without detectable IgG. The titers measured are very low, so it would be helpful to have data from negative control serum samples to demonstrate background of the assay and validate that the responses measured are real. It would also be informative to understand the neutralizing antibody responses at earlier timepoints closer to PSO, as perhaps they would be more robust. Likewise, it would be helpful to understand the titers of neutralizing antibodies measured in mild patients that do develop detectable IgG titers (Table 2) for comparison.

Data on the NAb titers of the negative controls in this assay has been added to the methods section (page 7, lines 158-160). We agree that it would be informative to investigate the NAb kinetics closer. However, NAb analyses are work-intense and expensive and we do unfortunately not have the resources to do this right now, but funds are being applied for and such an in-depth study is planned.

5: Figure 1C should include labels to differentiate the two sets of scatter plots (i.e. Days 1-35 and >75 days).

The annotation of days 1-35 and >75 days was mistakenly omitted; this has now been rectified.

6: Table 2: Annotations for “b” and “c” are mixed up in table legend.

This is now corrected (page 11, line 246-247).

Reviewer #2

1: General: Timelines of antibody seroconversions are referred to frequently in the manuscript without context – e.g. “11 vs. 22 days”. Does this refer to post symptom onset? This should be clarified throughout.

This does refer to post symptom onset, which has now been clarified throughout.

2: Abstract: In the last sentence of the abstract, it is stated that “Our results…are vital for…. estimating the true infection prevalence in populations.” Is it true that antibody serology assays are being used to measure both seroprevalence AND infection prevalence? It is my understanding that RT-PCR is being used in most cases to measure infection prevalence.

Reference to infection prevalence has been removed from the abstract (page 2, line 48). 

3: Introduction: Literature on SARS-CoV-2 serology is constantly increasing and our thinking is changing by the day. It would be worthwhile to add more context to previous literature, including what has emerged in the past few weeks since submission if possible.

We have added a section in the introduction with more recent references reflecting the recent conflicting findings in this field (page 3, lines 69-73).

4: Methods: I could not find any supplemental information for this manuscript and thus no detailed protocols. The authors do refer to the Amanat et al paper for their SARS-CoV-2 ELISA and it appears that they made some modifications, but more details and a rationale for changing the conditions would be very helpful to the reader. It would be helpful to add more details for the RBD ELISA, such as incubation times, what positive controls were used, the readout (concentration determined by normalizing to a standard curve or OD) and how the threshold for a positive response was determined. There were also several modifications done compared to the published method by Amanat et al, and it would be helpful to provide a rationale for these differences (such as the increased coating concentration, differences in buffers/diluents, etc). It is also important to state whether the samples were also heat inactivated before the analysis of binding antibodies (either with the commercial assays or the RBD ELISA). Please state what positive controls were used in the RBD ELISA and how the threshold for a positive response was calculated. For the neutralization assay, which strain of SARS-CoV-2 was used, and what were the controls in the assay? How was positivity determine in the neutralization assay?

We have now updated the method section to give extensive details regarding the RBD ELISA (Page 5, lines 121-135). AUC was calculated from the ELISA curves using GraphPad Prism as in Amanant et al and many other publications.

Most of the differences are in the secondary antibodies used: the secondary anti-human-IgG used by Amanant is no longer produced by ThermoFisher and anti-IgA and IgM were not used by Amanant et al. We have used secondary antibodies from the same supplier (SouthernBiotech). Sera were not heat inactivated, but we did run comparisons and found no differences in titers induced by heat inactivation.

For the neutralization assay, all sera with antibody titers ≥4 (1/4 dilution) were considered neutralizing, demonstrating neutralization in both 1/2 and 1/4 dilutions, confirming the presence of antibodies with capacity to block infection. Sera (n=17) from patients and blood donors from before December 2019 were used as a negative control and no neutralization was detected in any of these samples (page 7, lines 158-160)..

5: Data could not be found in a repository. The description of data location is “All files are available from the Zenodo database (DOI:XXX)” so I was unable to find this data.

See Journal Requirements #3.

6: Table 2: the footnotes for b and c are switched.

This is now corrected.

7: Figure 3: 

What p values are denoted by the * and **? 

It would also be helpful to have a positivity cutoff line, similar to what you have in Figure 1 as well as an explanation of how the positivity threshold was calculated. What percentage of samples in the Mild and Severe infection categories would be positive? How is AUC calculated and can this be added to the methods/stats section? What is the readout of the RBD ELISA- concentration or OD? Is it possible that the low titers of IgA and IgM are due to competition with high levels of IgG for binding to the RBD?

P-value summaries are now added to the figure legends (page 9, line 211 and page 10, line 232-233).

We have now added a cutoff line in Fig 3 at 2SD over the negative control average AUC.

The readout of RBD ELISA is an absorbance read at several serum dilutions. From the dilution curve, the AUC (area under the curve) was calculated using GraphPad Prism.

It is possible that competition between IgG and IgA/IgM would impact the titer, but we think it is unlikely. If titers were equal and the only difference would be the amount of antibodies, then we should see binding of IgA at a subsaturating IgG dilution, which we do not. Further, at a saturating IgG dilution we see increased IgA binding with decreasing dilution, indicating available binding sites on RBD. 

Discussion: It is intriguing that the authors claim they did not observe a notable decline in antibody responses with the commercial assays. This should be discussed in further detail in the context of current literature on SARS-CoV-2. Would the same results be expected if a longitudinal analysis was conducted using the RBD ELISA or the neutralization assay? Also, on page 12 line 277, the authors refer to the increased sensitivity of their in-house ELISA. Is this more sensitive than the one published by Amanat et al? Did the various modifications make the ELISA more sensitive than the one published by Amanat et al? Or is the statement just meant to compare the RBD ELISA to the commercial serology assays? 

That is an interesting and important question. There are by now a few papers suggesting sustained levels of IgG antibodies few weeks after infection. We would expect that RBD ELISA and neutralization titers would follow the general trend observed with Architect and iFlash, however we have no data supporting this hypothesis, which will be the topic of future studies. This is now commented upon in the discussion (page 13, lines 308-311).

Reviewer #3

1: In abstract, there is no methods section.

A sentence regarding the method has been added to the abstract (page 2, line 31-33).

2: For the RBD in-house antibody assay, how to decide the cut-off should be stated in the methods.

We have amended the methods and figure 3 legend to indicate how the cut-off was calculated (page 6, line 135).

3: For neutralizing antibody assay, how to decide the cut-off should be clarified in the methods.

We have amended the methods section to indicate how the cut-off was established (page 7, lines 156-160).

4: In Line 167, a p-value is needed.

This has been added (page 9, line 201).

5: Missing data from patient2 in Table 3 and Table 3 should be presented as supplemental table.

As explained (page 10, line 222-224), only two of the three non-seroconverting patients were available for additional testing, thus we have no additional data from patient 2.

6: In Line 273-282, the argument about anti-RBD-IgA may not be true. One sample in the M group has a very low IgA (the dot is touching the 0.00 base line in Figure 3). The high sensitivity of in-house assay may come with high false positivity. IgA covid tests are known for their high false positivity.

The reviewer is right that IgA COVID tests are known for false positive. Of note, our 10 negative controls have all low IgA levels to RBD. It is also correct that several patients in the mild group are negative for IgA. With the cut-off of average AUC of negative +2SD still >50% of patients with mild disease and severe disease were over the cut-off. However, as a group, the IgG- patients had a statistically lower IgA reactivity compared to other groups. 

In addition to these responses to reviewer comments, we have discovered an error in the data analysis. One sample from a patient with mild disease had been incorrectly entered as >75 days, thus there is now one person less in the mild group in figures 1C and 3. This has had no impact on the statistical significances.

---

## [Decision Letter · Decision Letter 1]

5 Oct 2020

PONE-D-20-21112R1

Serum-IgG responses to SARS-CoV-2 after mild and severe COVID-19 infection and analysis of IgG non-responders

PLOS ONE

Dear Dr. Gisslen,

Thank you for submitting your manuscript to PLOS ONE. After careful consideration, we feel that it has merit but does not fully meet PLOS ONE’s publication criteria as it currently stands. Therefore, we invite you to submit a revised version of the manuscript that addresses the points raised during the review process.

We look forward to receiving your revised manuscript.

Kind regards,

Stephen R. Walsh, MDCM

Academic Editor

PLOS ONE

Additional Editor Comments (if provided):

Thank you for addressing the reviewers' comments. We believe the manuscript has been improved. Please note that Figure 3 appears to be mislabeled.

Reviewers' comments:

Reviewer's Responses to Questions

**Comments to the Author**

1. If the authors have adequately addressed your comments raised in a previous round of review and you feel that this manuscript is now acceptable for publication, you may indicate that here to bypass the “Comments to the Author” section, enter your conflict of interest statement in the “Confidential to Editor” section, and submit your "Accept" recommendation.

Reviewer #1: (No Response)

Reviewer #2: All comments have been addressed

Reviewer #3: All comments have been addressed

2. Is the manuscript technically sound, and do the data support the conclusions?

Reviewer #1: Yes

Reviewer #2: (No Response)

Reviewer #3: Yes

3. Has the statistical analysis been performed appropriately and rigorously? 

Reviewer #1: Yes

Reviewer #2: (No Response)

Reviewer #3: Yes

4. Have the authors made all data underlying the findings in their manuscript fully available?

Reviewer #1: Yes

Reviewer #2: (No Response)

Reviewer #3: Yes

5. Is the manuscript presented in an intelligible fashion and written in standard English?

Reviewer #1: Yes

Reviewer #2: (No Response)

Reviewer #3: Yes

6. Review Comments to the Author

Reviewer #1: Figure 3: The figure legends for IgA and IgM do not match the y-axis labels on the graphs. Are the y-axis labels incorrect or are the graphs between Figures 3B and 3C switched?

Reviewer #2: (No Response)

Reviewer #3: The response is appropriate and the revision has addressed all my concerns . Thank you for all the works!

7. PLOS authors have the option to publish the peer review history of their article (what does this mean?). If published, this will include your full peer review and any attached files.

Reviewer #1: No

Reviewer #2: No

Reviewer #3: No

---

## [Author Response · Author response to Decision Letter 1]

7 Oct 2020

Dear Editor,

We are pleased that our revision of the manuscript“Serum IgG responses to SARS-CoV-2 after mild and severe COVID-19 infection and analysis of IgG non-responders” was to the reviewers’ satisfaction. 

Regarding Reviewer #1, comment #6: The y-axis labels in Figure 3B+C were indeed incorrect and have now been remedied.

---

## [Editor Report · Decision Letter 2]

9 Oct 2020

Serum-IgG responses to SARS-CoV-2 after mild and severe COVID-19 infection and analysis of IgG non-responders

PONE-D-20-21112R2

Dear Dr. Gisslen,

We’re pleased to inform you that your manuscript has been judged scientifically suitable for publication and will be formally accepted for publication once it meets all outstanding technical requirements.

Kind regards,

Stephen R. Walsh, MDCM

Academic Editor

PLOS ONE
---

## [Editor Report · Acceptance letter]

13 Oct 2020

PONE-D-20-21112R2 

Serum-IgG responses to SARS-CoV-2 after mild and severe COVID-19 infection and analysis of IgG non-responders 

Dear Dr. Gisslen:

I'm pleased to inform you that your manuscript has been deemed suitable for publication in PLOS ONE. Congratulations! Your manuscript is now with our production department. 

Kind regards, 

on behalf of

Dr. Stephen R. Walsh 

Academic Editor

PLOS ONE